# PhyloString: A web server designed to identify, visualize, and evaluate functional relationships between orthologous protein groups across different phylogenetic lineages

Claudia Dorantes-Torres[1], Maricela Carrera-Reyna[1], Walter Santos[1], Rosana Sánchez-López[2], Enrique Merino[1]*

1 Department of Molecular Microbiology, Instituto de Biotecnología, Universidad Nacional Autónoma de México, Cuernavaca, Morelos, México, 2 Department of Plant Molecular Biology, Instituto de Biotecnología, Universidad Nacional Autónoma de México, Cuernavaca, Morelos, México

* enrique.merino@ibt.unam.mx

## Abstract

Proteins are biological units whose essence is defined by their functional relationships with other proteins or biomolecules such as RNA, DNA, lipids, or carbohydrates. These functions encompass enzymatic, structural, regulatory, or physical interaction roles. The STRING database (Nucleic Acids Research, 8 Jan 2021;49(D1): D605-12) provides an index that defines the functional interaction networks between proteins in model organisms. To facilitate the identification, visualization, and evaluation of potential functional networks across organisms from different phylogenetic lineages, we have developed PhyloString (https://biocomputo.ibt.unam.mx/phylostring/), a web server that utilizes the indices of the STRING database. PhyloString decomposes these functional networks into modules, representing cohesive units of proteins grouped based on their similarity of STRING values and the phylogenetic origins of their respective organisms. This study presents and thoroughly discusses examples of such functional networks and their modules identified using PhyloString.

## Introduction

Proteins are fundamental units of cells, playing crucial roles in growth, development, maintenance, and functionality by contributing to the structure, metabolism, regulation, and communication within cellular systems. Their essence, as cellular entities, is defined by their physical and functional interactions with other proteins and various types of biomolecules, including DNA or RNA molecules, lipids, carbohydrates, and cofactors. Numerous databases have been established to comprehensively compile and analyze the vast results generated from protein interaction studies. The creation and maintenance of these databases involve methodological and conceptual challenges such as utilizing standardized vocabularies and formats for protein and interaction descriptions, meticulous curation of information from scientific publications,

**Data Availability Statement:** All relevant data are within the paper and its Supporting Information files.

**Funding:** The author(s) received no specific funding for this work. C.D-T. is a Ph.D. student enrolled in the Programa de Doctorado en Ciencias Biomédicas, Universidad Nacional Autónoma de México (UNAM) and recipient of a CONACYT fellowship (218897).

**Competing interests:** The authors have declared that no competing interests exist.

and establishing a consistent approach for identifying the same protein across different databases. These challenges and a review of the most significant protein-protein interaction databases have been extensively discussed in [1–3].

Among the various databases of protein-protein interactions, the STRING database deserves special mention due to its inclusion of physical and functional interactions between sets of two or more proteins [4]. This functional interaction of proteins can arise through various instances, such as sharing the same substrate in a metabolic pathway or being components of the same protein complex [5].

STRING is a comprehensive database that integrates experimentally validated and computationally predicted protein-protein interactions. Its initial version was developed as a web application primarily used to identify functionally related genes in prokaryotic organisms by analyzing the conservation of the same genomic neighborhood across different genomes [6]. Over time, STRING has expanded to include additional sources of information, such as (i) genomic context analysis, which involves examining gene fusion events, conservation of genomic neighborhood, and phylogenetic co-occurrence profiles of orthologous gene pairs across different genomes; (ii) primary evidence obtained from experimental protein-protein interactions and gene co-expression experiments, meticulously curated from the scientific literature; (iii) manually curated pathway databases; and (iv) utilization of automatic literature mining techniques to identify genes that are co-mentioned within the abstracts of scientific publications. Afterward, this information is weighted by STRING based on the probabilities of proteins being present in the same metabolic pathway, as defined by the metabolic maps from the KEGG database (Kyoto Encyclopedia of Genes and Genomes) [7]. These weighted values are then integrated into a meaningful confidence score ranging from zero to 1,000, representing the strength of the physical or functional relationships for all protein interactions.

A fundamental characteristic of STRING is that the information collected for one organism is transferred to other organisms within its database [4, 8]. To achieve this, functional equivalence is assigned to orthologous genes based on the orthologous groups defined by the COG database [9]. Consequently, STRING provides a user-friendly web page that presents information on protein-protein interactions as a network. In this network, proteins (or their corresponding orthologous COG groups) serve as nodes (or vertices), while the edges represent the different types of evidence supporting the physical or functional protein-protein interactions.

From a System Biology perspective, considering the entire network of physical or functional interactions within a cell is crucial to comprehensively define a protein's function in a specific organism. This implies that while orthologous proteins may share similar physicochemical or enzymatic properties, their functions can vary depending on the presence or absence of their respective interaction partners, which contribute to their functional networks.

In this study, we have developed PhyloString (https://biocomputo.ibt.unam.mx/phylostring/), a web server that enables the analysis of protein functional modularity across different phylogenetic lineages in biological processes. PhyloString utilizes the functional relationships between proteins as defined in the STRING database [5]. By clustering proteins into modules based on their STRING scores and phylogenetic origins, PhyloString visually represents these modules using a heatmap. Additionally, PhyloString evaluates the statistical significance of protein enrichment within the modules.

This article presents representative examples of differential protein-protein functional relationship modules identified among phylogenetic lineages using PhyloString. These examples are thoroughly discussed, highlighting the importance of understanding protein functional modularity across different evolutionary lineages.

## Materials and methods

### Data sources

The STRING database v12.0 compiles information from 12,535 organisms, out of which 2,426 are categorized as core or reference organisms. Details about these organisms are structured into flat files, which were downloaded from its website at https://string-db.org/. These files included, among others: a) the orthologous groups and their proteins (COG.mappings.v12.0. txt); b) the name of the organisms and their taxonomic IDs (species.v12.0.txt); c) association scores between orthologous groups (COG.links.v12.0.txt); d) the full name of the organism´s lineage (fullnamelineage.dmp).

This version of STRING includes functional relationships for protein orthology groups found mainly in bacterial organisms (COG) and eukaryotic organisms (KOG). To simplify, in this article, we will use the term COG to refer to both COG and KOG.

### Statistical analysis and heatmap construction

We developed an R script (R version 4.3.0) to parse the data obtained from the STRING database (version v12.0). PhyloString facilitates assessing the distribution of biological functions, metabolic processes, structural elements, or regulatory complexes across diverse phylogenetic clades. To achieve this, PhyloString carries out several bioinformatics processes, as shown in Fig 1.

The analysis initiates by selecting a characteristic COG group as a seed query for the case study. Using this seed group, PhyloString identifies all functionally related COGs, surpassing a specified cut-off value according to the STRING index. This set of COGs thus selected are functionally related, either by participating in the same biological process, metabolic pathway, or structural complex, so they can be considered to form a network of functional protein-protein interactions (Fig 1A). Subsequently, PhyloString evaluates the presence (1) or absence (0) of each COG defined as functionally related to the seed group within various phylogenetic clades. PhyloString assigns a value of 0 to a COG if the number of organisms with that COG in that phylogenetic clade is less than 50%, while a value of 1 is assigned if the frequency exceeds this threshold (Fig 1B). These binary values are then multiplied by the corresponding STRING functional relationship index recorded for each COG pair (Fig 1C). The resulting values are employed to construct protein-protein interaction heatmaps (Fig 1D).

To depict the functional proximity between the initial seed COG and each COG presented in the heatmap, a color code was employed. The colors ranged from lighter shades like yellow to darker hues such as cherry or black, representing weak to strong functional associations, respectively. The value scale diagram used in all heatmaps is illustrated in S1 Fig.

PhyloString enables users to conduct this analysis at two taxonomic levels: the phylum and class levels. To ensure rapid performance, PhyloString provides a collection of *pre-processed analyses* with readily selectable default values from the main web page. Nevertheless, PhyloString also offers flexibility for *Advanced analysis*, allowing users to customize parameter values to their preferences.

Protein-protein interaction heatmaps are created using the *pheatmap* library in the RStudio software (Version 1.0.12). This library generates heatmaps with hierarchical clustering of the input data matrix applied to both rows, representing the different phylogenetic clades, and columns, representing the different orthologous groups (COGs).

Among the various distance metrics accessible in *pheatmap*, the Euclidean distance is the default choice in our pre-compiled analyses. This preference is due to its simplicity and capability to generate biologically meaningful outcomes in most cases. Furthermore, the Euclidean

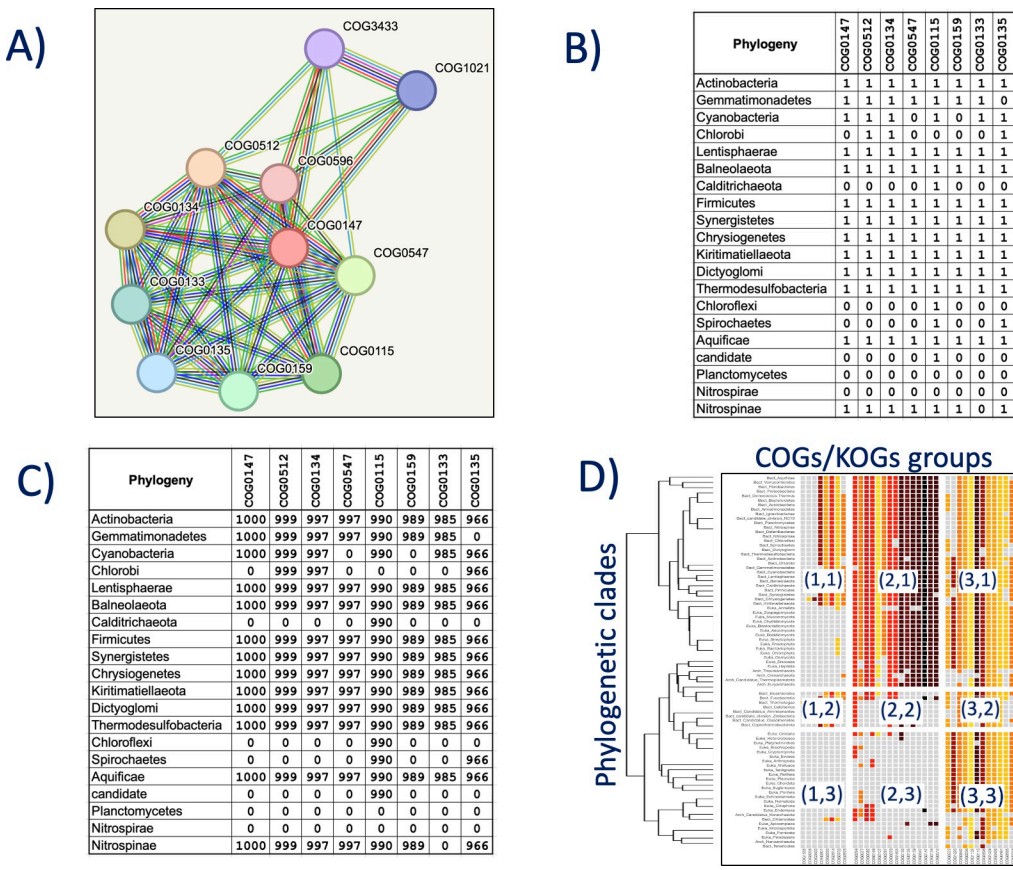

**Fig 1. Schematic representation of the primary steps employed by PhyloString.** A) Using a COG as a query seed that describes the biological functions, metabolic processes, structural elements, or regulatory complexes PhyloString selects the orthologous groups meeting the user-defined STRING cutoff value, forming a network of protein-protein functional interactions. B) PhyloString assesses the frequency of each COG presence within the functional interaction network across genomes in various phylogenetic clades. This information is used to construct a binary matrix indicating presence (1) or absence (0). A value of 0 is assigned if the number of organisms with a particular orthologous group in a phylogenetic clade is less than 50%, and 1 otherwise. C) Each cell in the binary presence/absence matrix is multiplied by the STRING index representing the functional association between the seed COG and the COG of that cell. D) PhyloString generates a heatmap using these values and applies hierarchical clustering to both rows (representing different phylogenetic clades) and columns (representing distinct COG orthologous groups). The clustering process results in the heatmap being divided into specific areas or modules, indicating cohesive units of highly functionally related proteins. In the article, these modules are denoted in parentheses by two numbers separated by a comma, representing the horizontal and vertical positions, respectively. The horizontal position corresponds to the COGs cluster number, and the vertical to the phylogenetic cluster number.

distance stands as one of the most frequently employed distance metrics in various biological studies. However, within the *Advanced analysis* option of PhyloString, users have the flexibility to opt, in addition to the Euclidean distance, other metrics, including the Manhattan distance, the Minkowski distance, and the Maximum distance. In addition, the new version of PhyloString includes new options for the clustering procedure other than the Complete method, which is the default method. In the *Advanced analysis* option of PhyloString, users can use alternative clustering methods, including the Single, Average, Median, Ward.D, Ward.D2, and Centroid methods. Descriptions of the Clustering methods and Distance matrices employed in the clustering procedure are described in the S1 and S2 Tables, respectively, and included in the new PhyloString Help webpage.

The heatmaps are divided horizontally and vertically based on the user-selected number of COG and phylogenetic divisions, respectively. These selections are then used as the *cluster_rows and cluster_columns* parameters in the *pheatmap* library. To evaluate the enrichment of COG groups within the various modules, we used the hypergeometric distribution, which models the probability of achieving a precise count of successes when making a set number of draws without replacement from a finite population composed of known successes and failures. The analysis of the hypergeometric distribution was conducted using the *phyper* and *dhyper* functions of the RStudio software and involves four key parameters, which are as follows:

1. The Population Size (N), in our case, corresponds to the total number of cells in the heatmap, which is determined by multiplying the number of phylogenetic clades by the number of COGs in the heatmap.

2. The number of Successes in the Population (H) represents instances where any COG within the heatmap is found in at least 50% of the organisms from a particular phylogenetic clade under study. In simpler terms, it counts all heatmap cells with values greater than 0 in this context.

3. The Sample Size (n) refers to the count of cells present in a specific subset of the heatmap. This subset is determined by the combination of COG groups and phylogenetic classes identified during the clustering process. The total number of potential heatmap subsets corresponds to the product of the chosen COG groups and phylogenetic groups initially selected by the user for analysis.

4. The number of Successes in the Sample (h) represents the count of cases in which any COG exists in at least 50% of the organisms within a particular phylogenetic clade present in the module of the heatmap under analysis. In simpler terms, it signifies all cells within the module of the heatmap with values greater than 0.

## Taxonomic classification

The taxonomic assignment of the organisms was carried out according to the KEGG GENOME database [7] (https://www.genome.jp/kegg/genome/).

## Defining COG seeds from protein sequences

As previously mentioned, every PhyloString analysis requires a COG seed as the central point around which functional neighbors are selected for heatmap display. In cases where there is no readily available COG seed or a keyword to facilitate seed identification, but a protein sequence is related to the study, PhyloString can determine its corresponding COG based on its sequence similarity to a reference database build with a non-redundant set of proteins of the STRING´s core organisms. To this end, the proteins of organisms classified as core were grouped in accordance with their COG assignation. The groups built in this way were processed to remove redundant proteins using the cd-hit program [10] and a sequence identity using a threshold of 0.8.

## Webpage construction

The PhyloString webpage comprises a four-layer structure: i) the database layer, where the data is processed by *ad hoc* scripts that read several input parameters and pre-process data from our set of 4,510 COG and 3,131 KOG groups; ii) the frontend layer comprises html5, JavaScript, and Cascading Style Sheets 3 (CSS3) codes. This second layer allows the users to

manipulate the backend layer; iii) the pre-render layer is an intermediate layer where the server accesses its APIs, retrieves values from the database, and establishes the foundation for creating charts based on user-input data. This is the most complex layer, constructed using a combination of PHP, JavaScript, Perl, and C frameworks, as well as custom scripts over an implementation of the amCharts software; iv) the backend layer is the last. It takes user input selections, processes them using allocated values from the database, and generates human-readable graphics presented to the user inside the frontend layer. This backend layer was written in Hypertext Preprocessor (PHP) and Perl program languages.

In addition to the primary PhyloString page, users can refer to the "Help" and "Tutorial" pages for guidance on the key steps involved in the study. These resources serve as helpful guides, offering advice on analysis approaches that can enhance the overall understanding of the platform. These "Help" and "Tutorial" pages can be access by their corresponding buttons at the PhyloString page. PhyloString webpage can be accessed via the following link: https://biocomputo.ibt.unam.mx/phylostring/

## Results and discussion

### Web page construction

We developed PhyloString, a web page designed to analyze protein-protein functional interaction networks based on data from the STRING database. These networks are represented as heatmaps, where each cell reflects the product of two factors: the binary classification of COG group frequencies in the studied phylogenetic lineages and the STRING score between pairs of COG groups. To perform a quick inspection, PhyloString has implemented a function that allows the user to obtain the information of each cell within the heatmap when the user hovers their mouse over it. This information includes the phylogenetic group, the COG ID with its description, the count of organisms considered in the specific cell, and its corresponding score.

To assess the prevalence of specific subsets of COGs within protein-protein functional interaction networks, we performed a clustering analysis of the heatmap cells using their corresponding STRING values. Our approach involved dividing the list of COGs and organisms into user-defined divisions, COG centers (2 to 6), and Phylogenetic centers (3 to 5), as displayed on the PhyloString webpage. As a result, the heatmaps are segmented into distinct rectangular areas called "Modules." These modules represent cohesive sets of proteins that perform closely related functions within the entire functional network. Furthermore, PhyloString evaluates the statistical significance of COG enrichment within different modules of the heatmaps, presenting the enrichment analysis results as supplementary tables.

To facilitate the functional networks analysis, PhyloString provides flexibility in setting cut-off values for STRING scores (specifically, 900, 850, 800, 750, and 700). Additionally, users can choose the desired level of taxonomic depth for analysis, whether by considering the *Phylum* or *Class* of the organisms. This customizable approach enables users to explore and interpret the data more effectively.

PhyloString offers a help webpage and a tutorial webpage designed to offer users a comprehensive understanding of the web server. These pages aim to clarify the necessary input parameters for their studies, outline the different sections in the analysis output, and guide users through the essential steps of a standard study. Additionally, they provide valuable tips for maximizing the capabilities of PhyloString. The key steps utilized by PhyloString are depicted in Fig 1.

## Representative examples of the use of PhyloString

To illustrate the diverse range of analysis achievable with PhyloString, we present a set of examples centered in essential cellular processes. The functional relationships between proteins in these processes are visualized as heatmaps, where cohesive and integrated modules are identified, representing tightly functionally related proteins. Throughout the text, the modules are denoted by their positions in the heatmaps, enclosed in round parentheses, representing their corresponding COG and phylogenetic groups, respectively.

**i) Ribosomal and other proteins that are involved in the translation process.** Ribosomes are essential cellular complexes composed of small proteins and RNA molecules. They play a crucial role across all three domains of life, Bacteria, Archaea, and Eukarya, in translating mRNA into proteins by catalyzing the assembly of amino acids into polypeptide chains. However, the complexity of ribosomes varies depending on the phylogenetic origin of the organism. Bacterial organisms show the lowest, followed by archaeal organisms, while eukaryotic organisms exhibit the highest level of complexity in their ribosomal composition [11, 12].

Despite the differences in composition, ribosomes of all three domains of life possess a common set of 34 proteins that are universally conserved and are the central core of their structure. This set of proteins is responsible for the ribosome´s main functions, such as peptide bond formation during protein synthesis [11, 12]. In addition, there is a set of ribosomal proteins common to archaeal and eukaryotic organisms, while some others are exclusively found in bacterial and eukaryotic organisms.

To investigate the phylogenetic distribution of proteins involved in the translation process, we utilized PhyloString to analyze the functional relationships within the COG0199, which clusters the S14 orthologous ribosomal proteins. The resulting analysis is presented in the heatmap shown in Fig 2. This figure reveals that COGs of modules (4,1), (4,2), and (4,3) (COG group 4, S3 Table) form part of the core ribosomal proteins shared by Bacteria, Archaea, and Eukarya (phylogenetic groups 1–3, S4 Table). The statistical values for the COG enrichment within this group can be found in the supplementary material (S5 Table). Furthermore, we observed that COGs of modules (3,2) and (3,3) mainly correspond to proteins shared by Archaea and Eukarya, while COGs of modules (2,1) and (2,3) are primarily associated with proteins shared by Bacteria and Eukarya. Lastly, module (1,1) corresponds to proteins mainly present in Bacteria.

**ii) Amino acid biosynthesis. Tryptophan autotrophic and heterotrophic organisms identified by PhyloString.** L-tryptophan is one of the scarcest amino acids found in most proteins and the most costly to synthesize in terms of cell resources [13]. In many bacterial organisms, the genes coding for the enzymes involved in the tryptophan biosynthesis are transcribed in the same polycistronic unit to facilitate their coordinated expression. To identify the overall distribution of L-tryptophan autotrophic and heterotrophic organisms, we used the COG0147 that clustered the anthranilate synthases component I enzymes, TrpE. The result of this analysis is shown in Fig 3.

Module (2,1) of the heatmap of Fig 3 clusters all the autotrophic organisms having the enzymes involved in L-tryptophan biosynthesis, which are clustered COG group 2 (S6 Table), and include the following enzymes: Anthranilate synthase TrpE (COG0147), which forms a complex with TrpG (COG0512) to convert chorismate and glutamine into anthranilate [11]; TrpD (COG0547), known as anthranilate phosphoribosyl transferase, which transfers the phosphoribosyl group from 5-phosphorylribose-1-pyrophosphate to anthranilate, resulting in phosphoribosyl anthranilate; TrpF (COG0135), or phosphoribosyl anthranilate isomerase, which converts the phosphoribosyl anthranilate into carboxyphenylamino-deoxyribulse-5-phosphate. Additionally, TrpC (COG0134), referred to as indol-3-glycerol phosphate

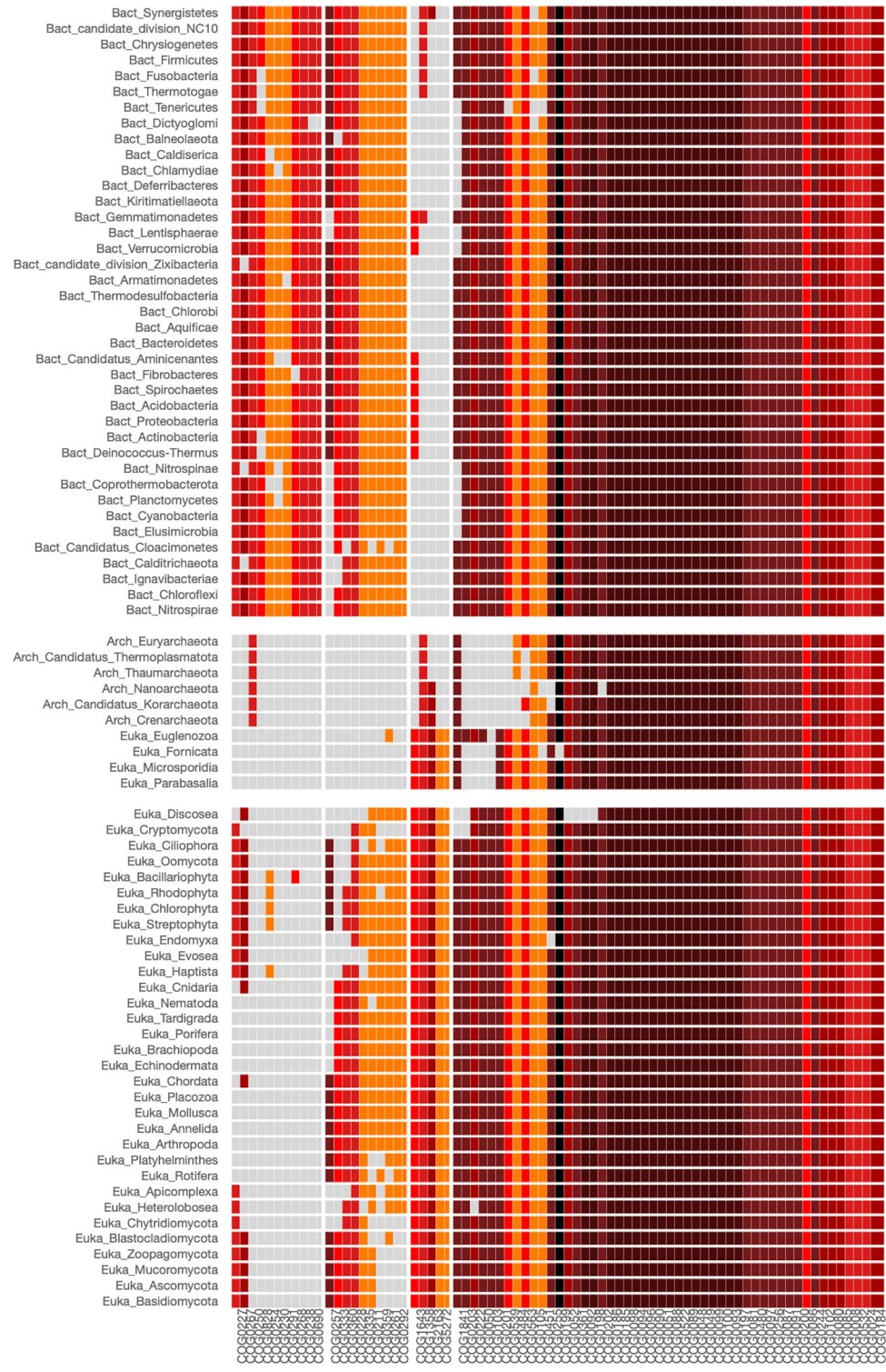

**Fig 2. Heatmap of COG0199 involved in the translation of protein process.** Ribosomal protein S14 orthologous belongs to this COG0199. The parameters used in the analysis were: Phylogenetic level: Phylum, STRING cut-off: 850, COG centers: 4, Phylogenetic centers: 3. Conserved COGs found across all three kingdoms of life, showing widespread distribution, are often crucial in the translation protein process. As a result, they typically appear in dark red in color, with STRING index values approaching the maximum of 1,000.

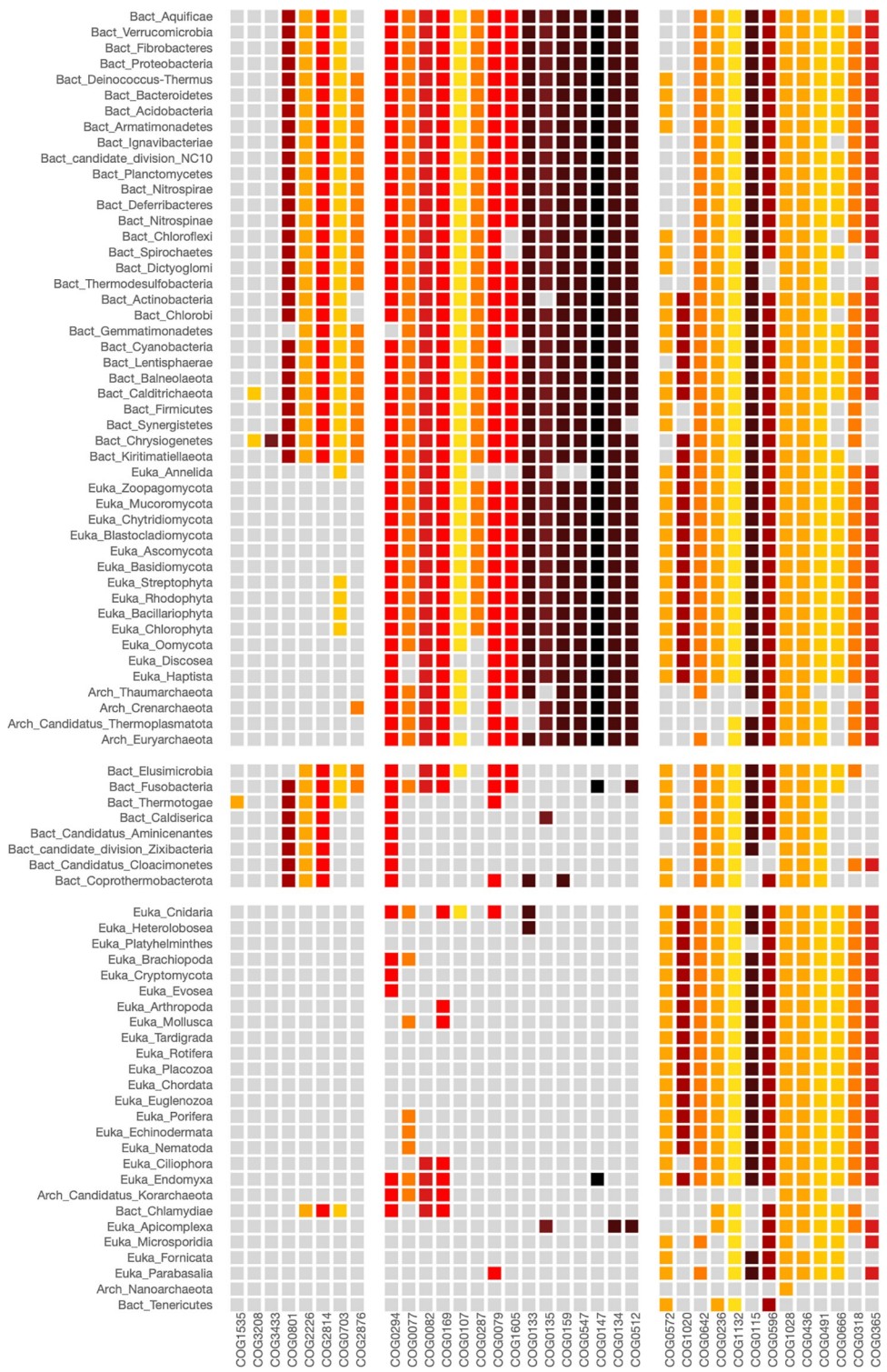

**Fig 3. Heatmap of COG0147 involved in tryptophan biosynthesis.** The COG0147 clusters the Anthranilate synthases component I enzymes, TrpE. The parameters used in the analysis were: Phylogenetic level: Phylum, STRING cut-off: 800, COG centers: 3, Phylogenetic centers: 3. COGs that are crucial in the biosynthesis of the amino acid tryptophan are predominantly depicted with a dark red color, indicating STRING index values nearing the maximum of 1,000. The first phylogenetic group comprises phyla whose organisms are typically heterotrophic with respect to tryptophan.

synthase, removes the carbon dioxide from carboxyphenylamino-deoxyribulse-5-phosphate resulting in the formation of indole-3-glycerol phosphate. Finally, the tryptophan synthase TrpA alpha chain (COG0159) cleaves the indoleglycerol phosphate to generate glyceraldehyde 3-phosphate and indole. The indole molecule generated in the previous step is used with L-serine by TrpB (COG0133), the beta chain of tryptophan synthase, to synthesize L-tryptophan.

Although the biosynthesis of L-tryptophan from chorismate commonly requires the presence of all the pathway's enzymes, in exceptional cases, the absence of an enzyme could be complemented by enzymes with a related activity. Notably, this complementation has been documented in the case of Actinobacteria that have lost the phosphoribosyl anthranilate isomerase *trpF* gene (COG0135), which has been complemented by the enzyme PirA that provides the required isomerase activity [14]. Using PyloString, we identified that analogous complementation could also have occurred in some organisms that are members of the Thaumarchaeota phylum (see module (2,1)). Furthermore, the absence of certain enzymes within the modules (2,1) could indicate that certain organisms are in the process of losing the L-Trp pathway, as could be the case of the organisms belonging to the Annelida phylum. This phenomenon has been observed in various pathogens, including certain species within the genera Haemophilus, Coxiella, and Propionibacterium [15].

Organisms of the phylogenetic phyla of the module (2,1) of Fig 3 are considered prototrophs and are predominantly bacteria (Phyla group 1, S7 Table). On the other hand, organisms of modules (2,2), and (2,3) comprise auxotrophic organisms that require external sources of tryptophan for their growth and survival. This group of organisms primarily includes eukaryotic organisms (groups 2 and 3, S7 Table). The under or over-representation of the COGs within the different modules for the heatmap of COG0147 is presented in the S8 Table.

To further investigate the phylogenetic distribution of L-tryptophan auxotroph organisms, we repeated a similar analysis by selecting the "Class" option in the Phylogenetic level panel of the PhyloString web page. The heatmap and corresponding Tables are presented as Supplementary Material (S2 Fig and S9–S11 Tables). Interestingly, the COG groups obtained using both the "Phylum" and the "Class" options are almost the same (S6 and S9 Tables), indicating that the functional relationships of the proteins involved in L-tryptophan biosynthesis are well-conserved across different phylogenetic lineages. However, when dividing the Phyla into their corresponding phylogenetic classes, specific tendencies within smaller groups of organisms become apparent. For instance, the Tissierellia and Erysipelotrichia classes, despite belonging to the Firmicutes phylum, which commonly comprises prototrophic organisms, lack the main enzymes of the L-trp biosynthesis pathway. A similar scenario is observed in the Proteobacteria class of Oligoflexia (S2 Fig and S10 Table).

**iii) Key proteins involved in the eukaryotic transcription initiation process.** Transcription is the process by which the RNA polymerases (RNAPs) copy the information contained in the DNA into molecules of RNA. Although RNAPs in all three domains of life share a common evolutionary origin, important differences have been documented in the molecular mechanisms and proteins related to the transcription initiation process. In the case of bacterial RNAPs, the recognition of specific promoter sequences in DNA is facilitated by sigma factors. However, in the case of archaeal and eukaryotic RNAPs, sigma factors are absent. Instead, their recognition of promoter elements relies on different generalized transcription factors (GTFs) [16, 17]. Furthermore, in eukaryotes, multiple types of nuclear RNAPs are responsible for transcribing distinct types of RNA. These transcription processes involve additional elements such as enhancers, silencers, and chromatin modifications (review in [18]).

One of the key components of the eukaryotic transcription initiation complex is the TATA-binding protein (TBP). Its primary function is to initiate transcription by RNA polymerase II at the promoters of protein-coding genes. TBP binds to a specific DNA sequence known as the

TATA box, typically found in the promoter region of genes. The binding of the TBP to DNA initiates the recruitment of other transcription factors and RNA polymerase II. TBP It is an integral part of a larger protein complex known as the TFIID complex, which includes various TBP-associated factors.

To have a broad overview of the proteins related to the eukaryotic transcription initiation process, we used the PhyloString web server with KOG1932 as input. This KOG corresponds to a TATA-binding protein-associated factor. The heatmap resulting from this analysis is shown in Fig 4, while the corresponding lists of COG groups and phylogeny groups, along with the statistical values indicating COG enrichment within this group, are provided in S12–S14 Tables, respectively.

Modules (1,1), (1,2), (1,3), and (1,5) in the heatmap shown in Fig 4 illustrate the presence of essential proteins involved in the RNA polymerase II transcription initiation. In contrast, modules (2,1), (2,2), and (2,3) cluster proteins with less significant relationships. These proteins are linked to various functions, including interactions with vitamin receptors, histone acetyltransferases, and proteins with annotations as "uncharacterized conserved protein".

Remarkably, our PhyloString analysis, using KOG1932 as a seed, uncovered that some clades of eukaryotic organisms lack most of the proteins associated with the canonical eukaryotic transcription initiation process. These organisms are positioned near the root of the phylogenetic tree that divides Bacteria, Archaea, and Eukarya. The most striking instances are the Fornicata and Euglenozoa phyla, which represent the closest lineages to the Last Eukaryotic Common Ancestor (LECA, S3 Fig). In the heatmap of Fig 4, module (1,4), Fornicata and Euglenozoa cluster with the bacterial phyla (S11 Table) since they lack most of the KOGs related to transcription initiation that are present in eukaryotic organisms. Additional examples of primitive eukaryotic phyla that lack important KOGs associated with transcription initiation, as identified by PhyloString, include Apicomplexa, Ciliophora, Bacillariophyta, and Parabasalia. These phyla are clustered in the phylogenetic group 5 (S11 Table) and the module (1,5) of the Fig 4.

Examples of lesser striking instances where KOGs involved in transcription initiation are absent can be observed in the Microsporidia, Haptista, Heterolobosea, and Rhodophyta phyla, which lack certain subunits of the transcription initiation factors. These phyla cluster together in the second phylogenetic group of Fig 4 and S13 Table.

**iv) Photosynthesis.** Photosynthesis is a vital biological process that many organisms employ to convert light energy into chemical energy and store it in organic compounds that can later be metabolized through cellular respiration to provide the energy required to perform different biological activities.

To have an overview of the functional network of proteins involved in this process, we used the COG5701, designated as Photosystem I reaction center subunit A1, PsaA, as input of our PhyloString web server. The corresponding heatmap is shown in Fig 5.

Remarkably, phylogenetic group 1 of the heatmap of Fig 5 exclusively comprises the Cyanobacteria phylum. These are photosynthetic bacteria that utilize chlorophyll and various pigments for photosynthesis. Unlike eukaryotic algae, they lack chloroplasts and instead perform photosynthesis within specialized cellular membrane structures. Furthermore, many cyanobacteria have the remarkable ability to capture atmospheric nitrogen. Their distinct blue-green color results from the presence of pigments like phycocyanin and chlorophyll, which overshadow the green tint of chlorophyll, providing cyanobacteria with their distinctive appearance [19, 20].

Phylogenetic group 2 of the heatmap of Fig 5 includes various phyla within the monophyletic group known as Viridiplantae, commonly referred to as green plants. These organisms get their green color from pigments like chlorophyll. Viridiplantae is characterized by its

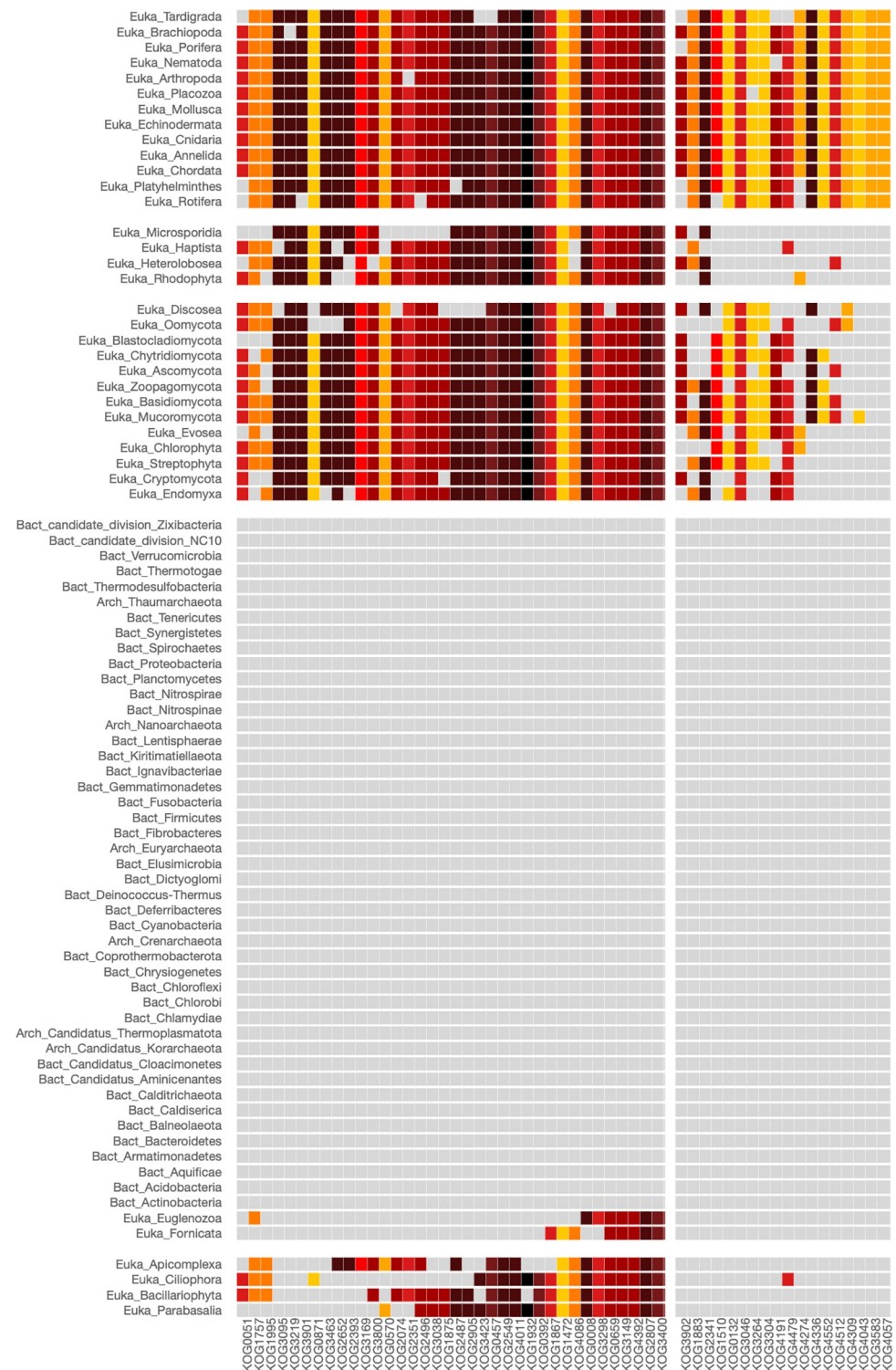

**Fig 4. Heatmap of KOG1932 involved in the transcription initiation process.** The KOG1932 corresponds to a TATA-binding protein-associated factor. The parameters used in the analysis were: Phylogenetic level: Phylum, STRING cut-off: 800, COG centers: 2, Phylogenetic centers: 5. The first COG group includes proteins crucial to the process of transcription initiation in eukaryotic organisms. As anticipated, these proteins are absent in the fourth phylogenetic group, primarily composed of Bacteria and Archaea. This block also encompasses the phyla Euglenozoa and Fornicata, known for being among the more ancient phyla within Eukarya.

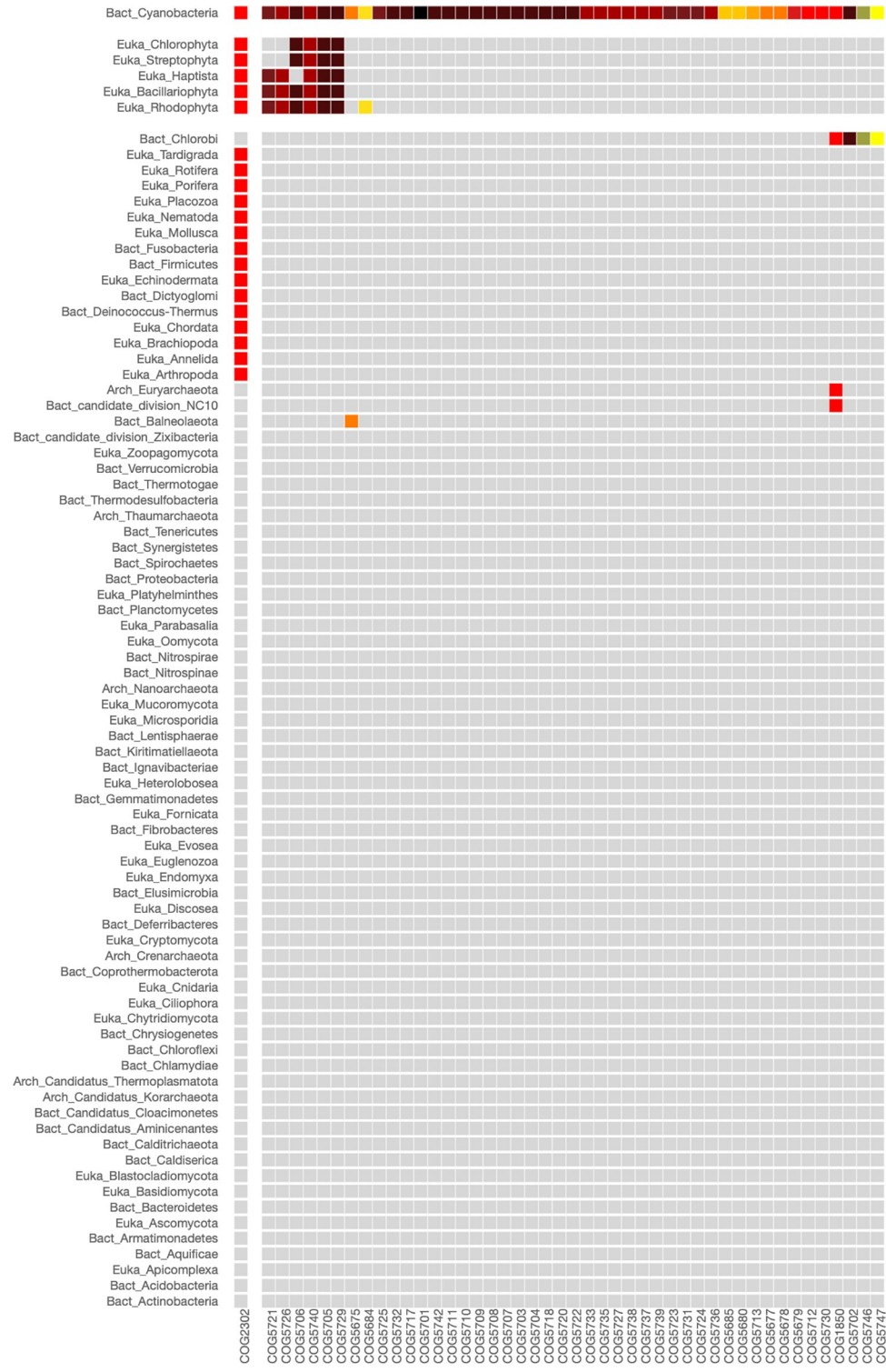

**Fig 5. Heatmap of COG5701 involved in the photosynthesis process.** The COG5701 corresponds to the photosystem I reaction center subunit A1, PsaA. The parameters used in the analysis were: Phylogenetic level: Phylum, STRING cut-off: 700, COG centers: 2, Phylogenetic centers: 3. Cyanobacteria, found in phylogenetic group 1, and specific eukaryotic phyla commonly recognized as green plants, possess the capability to carry out photosynthesis. This shared ability is reflected in the heatmap by the presence of essential orthologous proteins, represented as dark red cells.

members being photosynthetic and having chloroplasts, the cellular organelles responsible for photosynthesis. Viridiplantae is a highly diverse group, including green algae, land plants (Embryophyta), and their closest algal relatives (Charophyta). This diversity spans from unicellular green algae to complex, multicellular land plants, such as mosses, ferns, conifers, and flowering plants. Viridiplantae can be found in various habitats, including freshwater and marine environments, moist soil, and terrestrial ecosystems. Green algae in this group are typically aquatic, while land plants have adapted to life on land [21].

Interestingly, most of the COGs in the heatmap of Fig 5 are clustered in the second COGs group and are related to the different Photosystem reaction center proteins I and II, and cytochrome b6f complex subunit (S15 Table). The Photosystem II reaction center is a crucial component of the photosynthetic process in plants, algae, and cyanobacteria. Its role is to initiate the light-dependent reactions of photosynthesis. The heatmap of Fig 5 shows that a core of this set of enzymes is almost exclusively found in organisms of the Cyanobacteria phylum, which forms the first phylogenetic group. This core is also found in members of the second phylogenetic group, which includes various phyla of Viridiplantae, such as Chlorophyta, Streptophyta, Haptista, Bacillariophyta, and Rhodophyta (S16 Table). The values indicating COG enrichment within this group are detailed in S17 Table.

**v) Endocytosis and intracellular vesicle trafficking.** Conventional clathrin-mediated endocytosis and intracellular vesicle trafficking are essential and highly conserved cellular mechanisms in eukaryotes. They are responsible for transporting proteins and lipids either from the plasma membrane or their site of biosynthesis to their respective functional locations within the cell. Through endocytosis, the cell takes solutes (cargo) and fluids from the external environment. It also plays a key role in regulating the signaling cascade mediated by plasma membrane receptors; it is the route of entry for some pathogens, among other functions [22, 23]. On the other hand, intracellular vesicle trafficking is responsible for the molecular transport between endomembrane-based organelles, such as endoplasmic reticulum, Golgi complex, endosome, and vacuole/lysosome. Additionally, it is involved in exocytosis and the recycling of endosomes back to the plasma membrane. These processes are vital for maintaining cellular organization and functionality [24, 25].

Vesicles are formed by the cytosolic-oriented bending of the donor plasma membrane, which is accompanied by the assembly of a cage-like structure on the cytosolic side of the nascent vesicle. This cage-like structure can be composed of either clathrin light and heavy chains in clathrin-coated vesicles (CCVs) or the coatmer protein complex I and II in COPI and COPII vesicles, respectively [26–29].

In addition to cargo molecules, the membrane bending site recruits various accessory proteins, including Adaptor proteins (AP), cargo-receptors, ADP-ribosylation factor (Arf) small GTPases, RabGTPases, their specific guanine-exchange factors (GEFs), and activators (GAPs), as well as VAMP (vesicle-associated membrane proteins), SNARE proteins, among others [30–33]. Following the vesicle scission from the donor membrane, the cage-like structure disassembles. Afterwards, vesicle membrane fuses to the acceptor membrane. The membrane fusion process is facilitated by accessory proteins, including tethering proteins and SNAREs [34–38].

According to Thattai M. [39], there are "budding specificity " and "fusion specificity" modules comprising coat proteins and the corresponding paralogous variants of accessory proteins acting at different locations, respectively. Recent genetic findings on the First and Last eukaryotic common ancestor (FECA and LECA) provide information on the origin and evolution of organelles and the molecular and cellular mechanisms involved in intracellular vesicle trafficking (reviewed by [33, 40]).

We employed the PhyloString web server to explore clathrin-mediated endocytosis and vesicular trafficking among different phylogenetic groups. In this case, we employed KOG0985 (Vesicle coat protein clathrin, heavy chain) as the query seed KOG. The result of our analysis is presented as the heatmap of Fig 6 and Table 1, while the corresponding lists of COG groups and phylogeny groups, along with the statistical values indicating COG enrichment within this group, are provided in S18–S20 Tables, respectively.

As anticipated, structural, and accessory proteins related to clathrin-coated vesicles are exclusively grouped within the Eukarya phyla. These proteins include KOG0985 and KOG4031 (corresponding to vesicle coat protein clathrin's heavy and light chains, respectively), KOG0934, KOG0935, and KOG0936 (AP/clathrin adaptor complex, small subunits), along with other components linked to the clathrin-mediated endocytosis machinery (see Fig 6 and Table 1). Remarkably, the first phylogenetic group in the heatmap of Fig 6 reveals that the eukaryotic phyla Fornicata and Microsporidia are grouped with bacterial and archaeal phyla that lack clathrin-mediated endocytosis and intracellular vesicle trafficking. Consequently, related KOGs are absent in this grouping. Fornicata and Microsporidia phyla exhibit a limited number of KOGs, including KOG0985 (Vesicle coat protein clathrin), KOG0934 (Clathrin adaptor complex), KOG0935 (Clathrin adaptor complex), as well as additional adaptor proteins such as those of KOG1061 (Vesicle coat complex AP-1/AP-2/AP-4, beta subunit), KOG0937 and KOG0938 (members of the Adaptor complexes medium subunit family). Surprisingly, KOG4031 (vesicle coat protein clathrin, light chain) is absent in the Fornicata and Microsporidia phyla, as observed in the Parabasalia and Euglena phyla of the third phylogenetic group. The clathrin light chain is crucial for clathrin-coated vesicle (CCVs) formation, playing a key role in polymerization and serving as a connecting link with the actin cytoskeleton [27, 41]. The absence of KOG4031 may suggest the formation of CCVs with different properties or efficiencies. The second phylogenetic group in the heatmap of Fig 6 outlines KOGs present in metazoan, fungi, and amebozoan. It includes clathrin-mediated endocytosis and intracellular vesicle trafficking, as well as components somewhat associated with specific functional or cellular processes related to endocytosis (Fig 6 and Table 1). Conversely, the third phylogenetic group mainly consists of unicellular and pluricellular algae, microalgae, land plants, and protozoa. This group clusters a reduced set of KOGs related to structural components of CCVs and intracellular vesicles.

As KOGs related to COPI (KOG0292, Vesicle coat complex COPI, alpha subunit) and COPII (KOG1986, Vesicle coat complex COPII, subunit SEC23, and KOG0307, Vesicle coat complex COPII, subunit SEC31) vesicles were also observed in the heatmap of Fig 6, we conducted a new PhyloString analysis, utilizing KOG1078 (Vesicle coat complex COPI, gamma subunit) as query seed to find the phylogenetic distribution of other KOGs of proteins that are part of the Vesicle coat complex. The outcome of this analysis is illustrated in the heatmap of Fig 7, while the corresponding lists of COG groups and phylogeny groups, along with the statistical values indicating COG enrichment within this group, are provided in S21–S23 Tables, respectively.

Fig 7 reveals a set of KOGs involved in vesicle trafficking, including subunits of the Vesicle coat complex COPI (one alpha—KOG0292, two beta—KOG1058 and KOG0276, one epsilon —KOG3081, one gamma—KOG1078, and one zeta—KOG3343). These groups of orthologous proteins are consistently present across all Eukarya phyla. However, it is noteworthy that KOG1078 and KOG3343 are absent in the Fornicata phylum, as indicated in Fig 7 and Table 1.

Apart from the Vesicle coat complex COPI subunits highlighted in the Fig 7 heatmap, we have also identified the four structural subunits of the Vesicle coat complex COPII: Sec23 (KOG1986), Sec24 (KOG1985), Sec13 (KOG1332), and Sec31 (KOG0307), alongside the

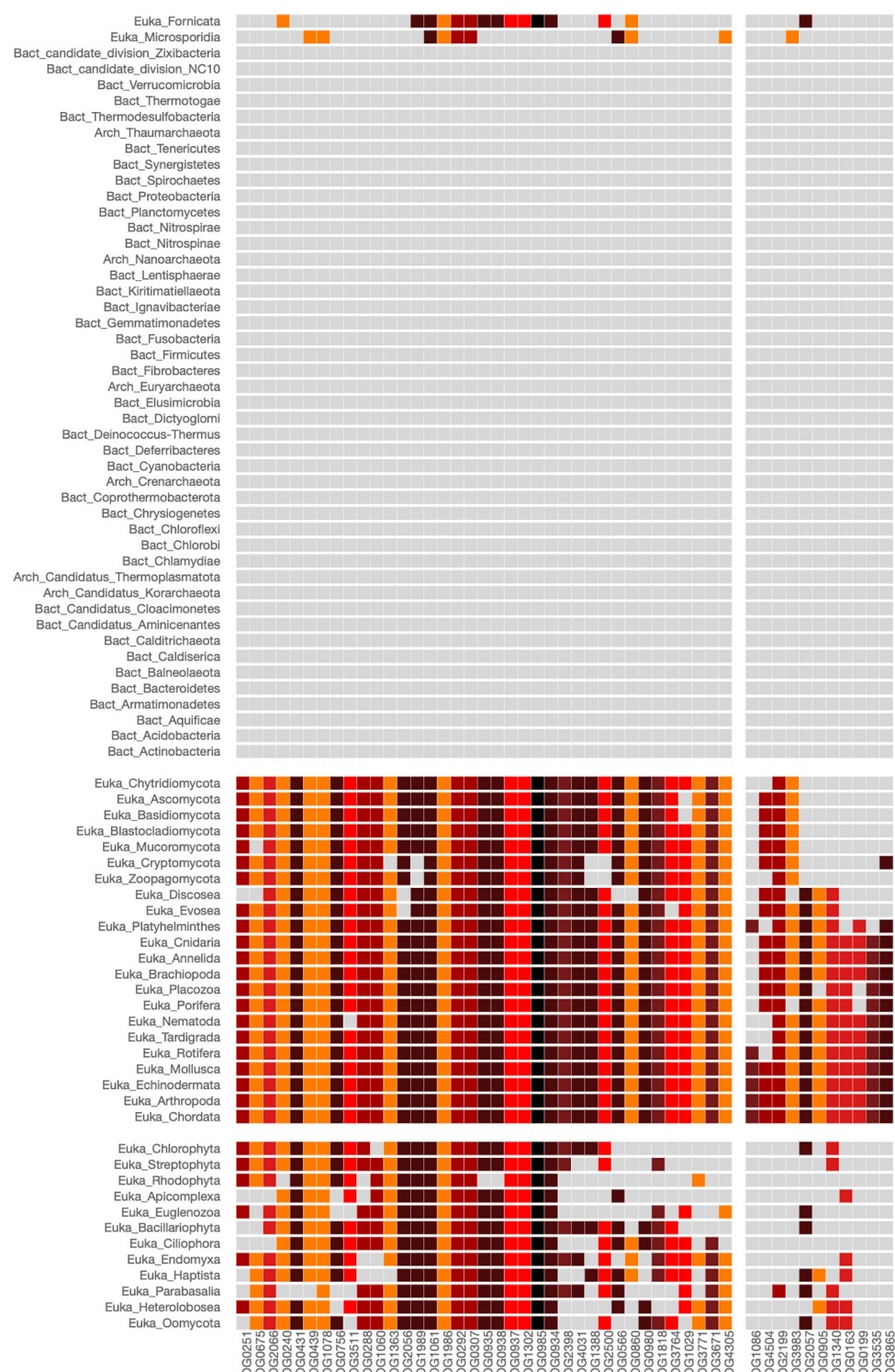

**Fig 6. Heatmap of KOG0985 involved in clathrin-mediated endocytosis.** The KOG0985 corresponds to the "Vesicle coat protein clathrin, heavy chain". The parameters used in the analysis were: Phylogenetic level: Phylum, STRING cut-off: 850, COG centers: 2, Phylogenetic centers: 3. The first phylogenetic group is mainly composed of bacterial and archaea phyla; it also includes the eukaryotic phyla Fornicata and Microsporidia, which exhibit a notable deficiency in proteins related to the clathrin-mediated endocytosis process. The second phylogenetic group encompasses phyla with organisms having structural components of endocytic and intracellular vesicles. It also includes components somewhat associated with specific functional or cellular processes related to endocytosis. The third phylogenetic group is

predominantly composed of unicellular eukaryotic organisms such as algae and protozoa, along with land plants in the Streptophyta phylum. Organisms in this group generally possess a reduced set of structural components related to endocytic and intracellular vesicles.

COPII-specific GTPase subunit SAR1 (KOG0077). The presence of these components spans across all Eukarya phyla, as depicted in the Fig 7 heatmap and succinctly outlined in Table 1.

In summary, the heatmaps of Figs 6 and 7 disclosed fascinating information concerning organisms of the primitive eukaryotic group Fornicata, as exemplified by the diarrheal pathogen *Giardia intestinalis* [42]. These findings suggest that these organisms might possess relatively rudimentary clathrin-coated vesicle (CCV) and COPI-like vesicle systems. While this hypothesis requires experimental validation, it is consistent with the observed reduced subcellular organization observed in *Giardia trophozoites*.

**Table 1. Phylogenetic distribution of the KOGs related to the Clathrin-mediated endocytosis.** The presence (checkmarks) and absence (crosses) of the main KOGs involved in Clathrin-mediated endocytosis are indicated for the most important phylogenetic groups, as could be inferred from our PhyloString analysis.

| | Function | Fornicata | Parabasalia | Apicomplexa | Euglenozoa | Evosea | Other eukarya | Archaea | Bacteria |
|---|---|---|---|---|---|---|---|---|---|
| CCV | **KOG0985.** Vesicle coat protein clathrin, heavy chain | √ | √ | √ | √ | √ | √ | X | X |
| | **KOG4031.** Vesicle coat protein clathrin, light chain | X | √ | X | X | √ | √ | X | X |
| | **KOG0934.** AP/Clathrin adaptor complex, small subunit | √ | √ | √ | √ | √ | √ | X | X |
| | **KOG0935.** AP/Clathrin adaptor complex, small subunit | √ | √ | √ | √ | √ | √ | X | X |
| | **KOG0936.** AP/Clathrin adaptor complex, small subunit | X | X | X | √ | √ | √ | X | X |
| COP I | **KOG1078.** Vesicle coat complex COPI, gamma subunit | X | √ | √ | √ | √ | √ | X | X |
| | **KOG3343.** Vesicle coat complex COPI, zeta subunit | X | √ | √ | √ | √ | √ | X | X |
| COP II | **KOG1986.** Vesicle coat complex COPII, SEC23 | √ | √ | √ | √ | √ | √ | X | X |
| | **KOG1985.** Vesicle coat complex COPII, SEC24 | √ | √ | √ | √ | √ | √ | X | X |
| Vesicle trafficking | **KOG0929.** Guanine nucleotide exchange factor | √ | √ | X | √ | √ | √ | X | X |
| | **KOG3103.** YIPF/Rab GTPase interacting factor | X | √ | √ | √ | √ | √ | X | X |
| | **KOG1300.** Sintaxin-binding/Munc18-Sec1 | √ | √ | √ | √ | √ | √ | X | X |
| | **KOG0810.** Q-SNARE protein Syntaxin 1 and related proteins involved in vesicle fusion | √ | √ | √ | √ | √ | √ | X | X |
| | **KOG2678.** Q-SNARE/syntaxin | X | √ | √ | √ | √ | √ | X | X |
| | **KOG0859.** R-SNARE, Synaptobrevin/VAMP-like protein | X | √ | √ | √ | √ | √ | X | X |
| | **KOG0860.** R-SNARE, Synaptobrevin/VAMP-like protein | √ | X | X | X | √ | √ | X | X |
| | **KOG0861.** R-SNARE protein YKT6, synaptobrevin/VAMP superfamily | √ | √ | √ | √ | √ | √ | X | X |
| | **KOG3251.** SNAP receptor/SNARE /vesicle-associated membrane protein (VAMP) | X | X | √ | √ | √ | √ | X | X |
| | **KOG1585.** NSF required for vesicle fusion in vesicular transport, gamma-SNAP | X | √ | √ | √ | √ | √ | X | X |
| | **KOG3369.** Trafficking protein particle (TRAPP) complex subunit | √ | √ | √ | √ | √ | √ | X | X |
| | **KOG3316.** Transport protein particle (TRAPP) complex subunit 6 | X | X | √ | √ | √ | √ | X | X |

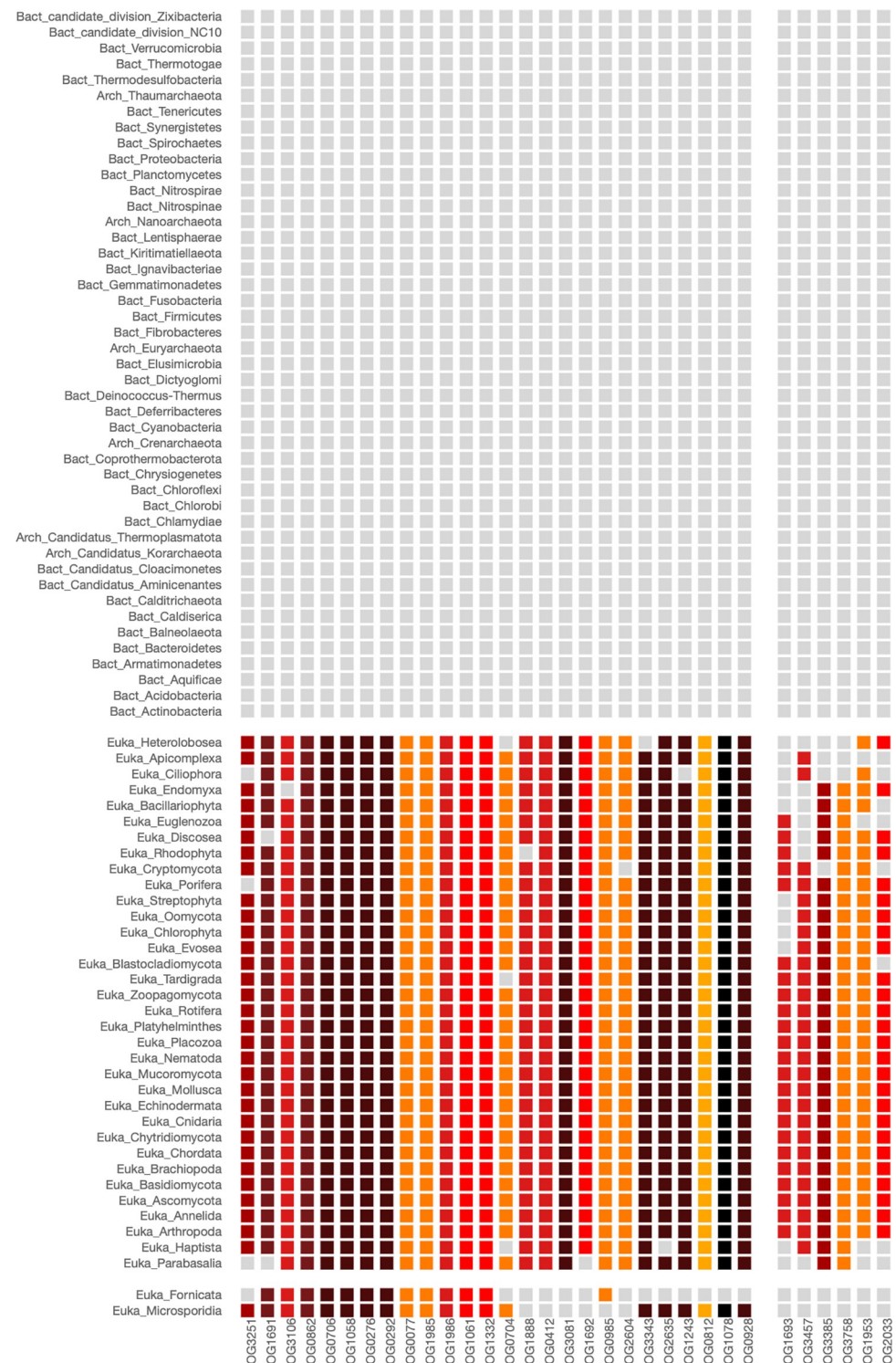

**Fig 7. Heatmap of KOG1078 involved in COPI-coated intracellular vesicles.** The KOG1078 corresponds to the "Vesicle coat complex COPI, gamma subunit". The parameters used in the analysis were: Phylogenetic level: Phylum, STRING cut-off: 850, COG centers: 2, Phylogenetic centers: 3. The first phylogenetic group is composed of Bacterial and Archaea phyla, which lack COPI intracellular vesicles. The second phylogenetic group includes Eukarya phyla. The phyla of this group is composed of organisms that possesses the structural components of COPI- and COPII-coated vesicles, as well as components somewhat associated to specific functional or cellular processes related to intracellular trafficking mediated by these vesicles. The third phylogenetic group is formed by the Fornicata and

Microsporida phyla. Organisms of these phyla have all the structural components of COPI- and COPII-coated vesicles, except for the epsilon, gamma and zeta subunits of the COPI complex (KOG3081, KOG1078 and KOG3343, respectively).

Giardia trophozoites exhibit a distinct endoplasmic reticulum network (ER) characterized by the presence of a unique vesicle system, known as peripheral vacuoles (PVs), positioned beneath the plasma membrane. This configuration may signify an evolutionary adaptation of the endocytic-exocytic pathways, potentially dependent on CCV- and COPI-like structures but lacking the clathrin light chain, as well as the COPI gamma and zeta subunits. Notably, PVs are contiguous with the ER but lack Golgi stacks [42, 43].

Giardia trophozoites exhibit a distinct endoplasmic reticulum network (ER) characterized by the presence of a unique vesicle system, known as peripheral vacuoles (PVs), positioned beneath the plasma membrane. This configuration may signify an evolutionary adaptation of the endocytic-exocytic pathways, potentially dependent on CCV- and COPI-like structures but lacking the clathrin light chain, as well as the COPI gamma and zeta subunits. Notably, PVs are contiguous with the ER but lack Golgi stacks [42, 43].

Typically, COPI vesicles play a crucial role in transporting cargo molecules between the Golgi apparatus and the ER, as well as within Golgi stacks [26]. The lack of certain components of the CCV and COPI machineries in other primitive Eukarya phyla, such as Parabasalia, Apicomplexa and Euglenozoa (Table 1) seem to be an indicative of the evolutionary events that contributed to the origin of multicellular organisms with a more complex cell biology, an avenue that would be interesting to explore.

## Conclusion

We have developed PhyloString, a valuable website that enhances the comprehension of functional relationships among protein sets and their phylogenetic prevalence across the three kingdoms of life. Through heatmap-based graphical representations of functional networks and statistical analysis, we could identify modules of proteins whose functions are tightly related. Our research presented a wide range of diverse and representative examples of various cellular processes that can be analyzed from both metabolic and evolutionary perspectives using our web server PhyloString.

## Supporting information

**S1 Fig. Scale diagram used to represent functional relations between COGs.** The colors ranged from lighter shades like yellow to darker hues like cherry or black, representing weak to strong functional associations.
(TIF)

**S2 Fig. Heatmap of COG0147 involved in tryptophan biosynthesis using class as Phylogenetic level.** The parameters used in the analysis were: Phylogenetic level: Class; STRING cutoff: 800; COG centers: 3; Phylogenetic centers: 3. The heatmap reveals that certain phylogenetic classes, even though they are part of a phylum typically associated with prototrophic organisms, do not possess the primary enzymes of the L-trp biosynthesis pathway. Examples include the classes Tissierellia, Erysipelotrichia, and Oligoflexia.
(TIF)

**S3 Fig. Phylogenetic tree of representative eukaryotic organisms.** The tree was drawn considering the phylogenetic distances reported for the iterative Tree of Life (iTOL) web page (https://itol.embl.de/). The name, phyla, and Tax ID of representative organisms are listed in

black, red, and blue, respectively.
(TIF)

**S1 Table. Clustering methods employed by PhyloString.** In the Customized analysis option, PhyloString provides the flexibility to utilize various grouping methods beyond the default "complete hierarchical" (agglomerative) method. These additional methods are briefly described in the Table.
(XLSX)

**S2 Table. Distance metrics used in the clustering procedure.** In the Customized analysis feature, PhyloString provides the flexibility to utilize alternative distance metrics alongside the default "Euclidean" distance. The distance metrics are briefly described in the Table.
(XLSX)

**S3 Table. COG groups of the heatmap of COG0199 involved in the protein translation process.** The parameters used in the analysis were: Phylogenetic level: Phylum; STRING cut-off: 850, COG centers: 4, Phylogenetic centers: 3.
(XLSX)

**S4 Table. Phylogeny groups of the heatmap of COG0199 involved in the protein translation process.** The parameters used in the analysis are the same as the S3 Table.
(XLSX)

**S5 Table. E-values of the heatmap of COG0199 involved in the protein translation process.** The parameters used in the analysis are the same as the S3 Table.
(XLSX)

**S6 Table. COG groups of the heatmap of COG0147 involved in the protein translation process, using Phylum as Phylogenetic level.** The parameters used in the analysis were: Phylogenetic level: Phylum, STRING cut-off: 800, COG centers: 3, Phylogenetic centers: 3.
(XLSX)

**S7 Table. Phylogeny groups of the heatmap of COG0147 involved in the protein translation process, using Phylum as Phylogenetic level.** The parameters used in the analysis are the same as in the S6 Table.
(XLSX)

**S8 Table. E-values of the heatmap of COG0147 involved in the protein translation process, using phylum as Phylogenetic level.** The parameters used in the analysis are the same as in the S6 Table.
(XLSX)

**S9 Table. COG groups of the heatmap of COG0147 involved in the protein translation process, using class as Phylogenetic level.** The parameters used in the analysis were: Phylogenetic level: Class, STRING cut-off: 800, COG centers: 3, Phylogenetic centers: 3.
(XLSX)

**S10 Table. Phylogeny groups of the heatmap of COG0147 involved in the protein translation process, using class as Phylogenetic level.** The parameters used in the analysis are the same as in the S9 Table.
(XLSX)

**S11 Table. E-values of the heatmap of COG0147 involved in the protein translation process, using class as Phylogenetic level.** The parameters used in the analysis are the same as in

the S9 Table.
(XLSX)

**S12 Table. COG groups of the heatmap of KOG1932 involved in the transcription initiation process.** The parameters used in the analysis were: Phylogenetic level: Phylum, STRING cut-off: 800, COG centers: 2, Phylogenetic centers: 5.
(XLSX)

**S13 Table. Phylogeny groups of the heatmap of KOG1932 involved in the transcription initiation process.** The parameters used in the analysis are the same as in the S12 Table.
(XLSX)

**S14 Table. E-values of the heatmap of KOG1932 involved in the transcription initiation process.** The parameters used in the analysis are the same as in the S12 Table.
(XLSX)

**S15 Table. COG groups of the heatmap of COG5701 involved in the photosynthesis process.** The parameters used in the analysis were: Phylogenetic level: Phylum, STRING cut-off: 700, COG centers: 2, Phylogenetic centers: 3.
(XLSX)

**S16 Table. Phylogeny groups of the heatmap of COG5701 involved in the photosynthesis process.** The parameters used in the analysis are the same as in the S15 Table.
(XLSX)

**S17 Table. E-values of the heatmap of COG5701 involved in the photosynthesis process.** The parameters used in the analysis are the same as in the S15 Table.
(XLSX)

**S18 Table. COG groups of the heatmap of KOG0985 involved in clathrin-mediated endocytosis.** The parameters used in the analysis were: Phylogenetic level: Phylum STRING cut-off: 850, COG centers: 2, Phylogenetic centers: 3.
(XLSX)

**S19 Table. Phylogeny groups of the heatmap of KOG0985 involved in clathrin-mediated endocytosis.** The parameters used in the analysis are the same as in the S18 Table.
(XLSX)

**S20 Table. E-values of the heatmap of KOG0985 involved in clathrin-mediated endocytosis.** The parameters used in the analysis are the same as in the S18 Table.
(XLSX)

**S21 Table. COG groups of the heatmap of KOG1078 involved in COPI-coated intracellular vesicles.** The parameters used in the analysis were: Phylogenetic level: Phylum STRING cut-off: 850, COG centers: 2, Phylogenetic centers: 3.
(XLSX)

**S22 Table. Phylogeny groups of the heatmap of KOG1078 involved in COPI-coated intracellular vesicles.** The parameters used in the analysis are the same as in the S21 Table.
(XLSX)

**S23 Table. E-values of the heatmap of KOG1078 involved in COPI-coated intracellular vesicles.** The parameters used in the analysis are the same as in the S21 Table.
(TXT)

## Acknowledgments

We sincerely thank Shirley Ainsworth for bibliographical services and Arturo Ocádiz and Juan Manuel Hurtado for their computer support.

## Author Contributions

**Conceptualization:** Claudia Dorantes-Torres, Rosana Sánchez-López, Enrique Merino.

**Data curation:** Claudia Dorantes-Torres, Rosana Sánchez-López, Enrique Merino.

**Formal analysis:** Claudia Dorantes-Torres, Maricela Carrera-Reyna, Walter Santos, Rosana Sánchez-López, Enrique Merino.

**Investigation:** Claudia Dorantes-Torres, Rosana Sánchez-López, Enrique Merino.

**Methodology:** Claudia Dorantes-Torres, Maricela Carrera-Reyna, Walter Santos, Enrique Merino.

**Resources:** Enrique Merino.

**Software:** Enrique Merino.

**Supervision:** Rosana Sánchez-López, Enrique Merino.

**Validation:** Enrique Merino.

**Visualization:** Maricela Carrera-Reyna, Walter Santos, Enrique Merino.

**Writing – original draft:** Claudia Dorantes-Torres, Rosana Sánchez-López, Enrique Merino.

**Writing – review & editing:** Claudia Dorantes-Torres, Rosana Sánchez-López, Enrique Merino.

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
