## [Decision Letter · Decision Letter 0]

6 Sep 2023

PONE-D-23-24307PhyloString: a web server designed to identify, visualize, and evaluate functional relationships between orthologous protein groups across different phylogenetic lineagesPLOS ONE

Dear Dr. Merino,

Thank you for submitting your manuscript to PLOS ONE. After careful consideration, we feel that it has merit but does not fully meet PLOS ONE’s publication criteria as it currently stands. Therefore, we invite you to submit a revised version of the manuscript that addresses the points raised during the review process.

We look forward to receiving your revised manuscript.

Kind regards,

Shailender Kumar Verma, Ph.D.

Academic Editor

PLOS ONE

Journal Requirements:

"C.D-T. is a Ph.D. student enrolled in the Programa de Doctorado en Ciencias Biomédicas, Universidad Nacional Autónoma de México (UNAM) and recipient of a CONACYT fellowship (218897)"

Reviewers' comments:

Reviewer's Responses to Questions

**Comments to the Author**

1. Is the manuscript technically sound, and do the data support the conclusions?

Reviewer #1: Yes

Reviewer #2: Partly

Reviewer #3: Yes

2. Has the statistical analysis been performed appropriately and rigorously? 

Reviewer #1: N/A

Reviewer #2: I Don't Know

Reviewer #3: N/A

3. Have the authors made all data underlying the findings in their manuscript fully available?

Reviewer #1: Yes

Reviewer #2: No

Reviewer #3: Yes

4. Is the manuscript presented in an intelligible fashion and written in standard English?

Reviewer #1: Yes

Reviewer #2: Yes

Reviewer #3: No

5. Review Comments to the Author

Reviewer #1: PhyloString is one of the most useful web application. While the content is ok but the website part needs a revision.

1. There is no help page.

2. One tutorial page that will walk through with an example run is missing in the design.

3. The example_seq looks like a nucleic acid sequence rather than a protein sequence. Although, I understand A, C, T, G codes for amino acids. Better change it to some protein sequence.

4. Contact information is missing in the website. In case of difficulty, how the users will contact the authors/developers?

5. You need to provide some example input and output.

Reviewer #2: The manuscript entitled « PhyloString: a web server designed to identify, visualize, and evaluate functional relationships between orthologous protein groups across different phylogenetic lineages. «  by Dorantes-Torres et al. reports on a web server for assessing and visualising potential functional networks across organisms from different phylogenetic lineages. They rely on public databases, mostly STRING and COG, and on existing methodologies to build heat maps and perform statistical enrichment analysis. The tool can be of interest for the community and the authors successfully showcase some meaningful examples. However, both the manuscript and the webserver could be largely improved.

Major concerns:

1/ The methods are not sufficiently described. Please provide some hints as to what the R functions used are actually doing. Which statistical test, clustering method, etc?

2/ The main result of the webserver is not clearly described. Could the authors explicitly write down the score that correspond to each cell in the heat map? In particular, it is not clear at all from the text that each cell correspond to some relationship between the protein in x-axis from the phyla in y-axis with the corresponding protein from the query COG. Moreover, do the author consider the presence/absence for the query only or for both proteins in the pair? In case a phylum contain several species, how is that dealt with to compute the score? How many species are there behind each heatmap?

3/ What is the pertinence of using a continuous color scale when the null values have a special meaning: absence of the protein (and not only of the interaction)?

4/ A lot of thresholds are used without justification. What would happen if we would vary the a bit? What would change if I change the query COG but I stay in the same functional set of proteins? How are the proteins in x-axis identified? Does the STRING index cutoff play a role in this selection? What would happen if I have 2 cells next to each other, one yellow and one red, and behind the yellow one the support for presence is 0.45 in the phylum, while it is 0.55 in the other cell? Is it really pertinent to put 0 and 1 which such similar proportions?

5/ The webserver lacks documentation. It is easy to use but it is not so easy to understand the output without any tutorial or description.

Minor points:

- the figure legends should be more detailed.

Reviewer #3: Dorantes-Torres et al. present a Web server, PhyloString, that analyses data, ingested from the STRING database and decomposes the associations in two-dimensional clustering, along the COG composition and the phylogenetic axes. The clustering is accompanied by a statistical enrichment/depletion estimate, based on an extremely naïve random independent combinatorial model.

The server exists and serves the information claimed by the authors; this, technically, gives the whole enterprise a passing score for publication. The utility of the analysis is doubtful, given the primitive statistics and numerous ad hoc decisions, implemented in the pipeline (e.g. the binary presence/absence scoring at 50% of occurrence among the genomes in a phylogroup). Using a hopelessly outdated COG annotation doesn't help either (e.g. CRISPR-Cas9 is annotated as "Uncharacterized protein conserved in bacteria", as it was in 2003 version). The authors make no attempt to find the data-driven granularity for clustering, leaving the user to experiment with the settings.

There is one problem that must be addressed in the revision: in the Methods section the authors mention using the R functions for hypergeometric distribution, but give no further details for the analysis they perform. Likewise, they use clustering facilities of the heatmap library without any reference to how the rows and columns are clustered. A detailed description of the analysis from the subject matter perspective (what is being computed and why, as opposed to details on how R functions are invoked), including the underlying assumptions and an assessment of their applicability to the real data, is necessary.

6. PLOS authors have the option to publish the peer review history of their article (what does this mean?). If published, this will include your full peer review and any attached files.

Reviewer #1: No

Reviewer #2: No

Reviewer #3: No

---

## [Author Response · Author response to Decision Letter 0]

24 Nov 2023

Rebuttal letter 

PONE-D-23-24307

Response to Reviewer #1: 

Reviewer #1: PhyloString is one of the most useful web application. While the content is ok but the website part needs a revision.

1. There is no help page with clear instructions.

We have introduced a Help web page that provides an overview of the PhyloString web server, explains the input parameters for the study, and outlines the various sections of the analysis output. Specifically, lines 258 to 264 in the revised version of our article highlight the availability of PhyloString's Help and Tutorial pages, serving as guides for users in navigating the main steps of their study.

2. One tutorial page that will walk through with an example run is missing in the design.

In addition to the Help page, our PhyloString web server now includes a Tutorial web page designed to guide users through the key steps of a standard study, along with valuable tips for maximizing the capabilities of PhyloString and effectively interpreting the diverse potential outcomes. Furthermore, the PhyloString Tutorial web page features illustrative examples, some of which are also presented in the article. As mentioned in the previous response, lines 258 to 264 in the revised version of our article emphasize the accessibility of PhyloString's Help and Tutorial pages.

3. The example_seq looks like a nucleic acid sequence rather than a protein sequence. Although, I understand A, C, T, G codes for amino acids. Better change it to some protein sequence.

Our updated version of PhyloString replaced the example sequence with an actual protein sequence. Furthermore, we have introduced a new feature in PhyloString that enables users to choose a COG/KOG using keywords that describe the functions associated with those orthologous groups.

4. Contact information is missing in the website. In case of difficulty, how the users will contact the authors/developers?

The contact information has been included at the bottom of the new version of our PhyloString web server.

5. You need to provide some example input and output.

As commented earlier, the Tutorial web page of the new version of our PhyloString website includes a set of practical examples, some of which are also showcased in the article. 

Response to Reviewer #2:

Reviewer #2: The manuscript entitled « PhyloString: a web server designed to identify, visualize, and evaluate functional relationships between orthologous protein groups across different phylogenetic lineages. « by Dorantes-Torres et al. reports on a web server for assessing and visualising potential functional networks across organisms from different phylogenetic lineages. They rely on public databases, mostly STRING and COG, and on existing methodologies to build heat maps and perform statistical enrichment analysis. The tool can be of interest for the community and the authors successfully showcase some meaningful examples. However, both the manuscript and the webserver could be largely improved.

Major concerns:

1/ The methods are not sufficiently described. Please provide some hints as to what the R functions used are actually doing. Which statistical test, clustering method, etc?

In the updated version of our article, lines 124 to 224, we have provided a more comprehensive explanation of R functions and the main variables used in the analysis. In addition, as indicated in lines 173 to 176, the new version of our PhyloString web server includes new options for the Clustering methods and Distance matrices that could be employed in the clustering procedure. The descriptions of these methods and distance metrics are presented in the new PhyloString Help webpage, as indicated in lines 187 to 196.

2/ The main result of the webserver is not clearly described. Could the authors explicitly write down the score that correspond to each cell in the heat map? In particular, it is not clear at all from the text that each cell correspond to some relationship between the protein in x-axis from the phyla in y-axis with the corresponding protein from the query COG. Moreover, do the author consider the presence/absence for the query only or for both proteins in the pair? In case a phylum contain several species, how is that dealt with to compute the score? How many species are there behind each heatmap?

 In response to the reviewer's feedback and to better describe “The main result of the web server,” we have included a new figure (Fig 1) to represent the primary steps employed by PhyloString schematically. In addition, we have improved the description of the main steps of the PhyloString analysis and how the heatmap is generated and could be interpreted. In addition, as indicated in lines 153 to 224. 

Furthermore, we have also implemented a novel approach for presenting the specific information for each cell within a heatmap. In contrast to the previous display of a basic HTML file, the enhanced version of PhyloString now offers an interactive feature that provides valuable information for each cell when the user hovers their mouse cursor over it. This information includes the phylogenetic group, the COG or KOG ID and its description, the number of organisms considered in the count for the specific cell, and its corresponding score. Details regarding implementing this functionality can be found in the revised article, lines 243 to 257.

3/ What is the pertinence of using a continuous color scale when the null values have a special meaning: absence of the protein (and not only of the interaction)?

The gradient color scale employed by PhyloString holds significant importance in understanding the potential impact of the absence of a protein or a set of related proteins (modules) within the studied functional interaction network. This significance is exemplified in the second instance discussed in our article, focusing on the biosynthesis of the amino acid tryptophan (Fig 3).

In this context, the absence of the enzyme phosphoribosyl anthranilate isomerase TrpF (COG0135) in the phylum Actinobacteria is readily interpretable due to the corresponding color approaching the highest possible value (1,000), indicating its considerable relevance in the biosynthetic process. Conversely, the yellow-like colors assigned to COG1535 (Isochorismate hydrolase), and COG3208 (Surfactin synthase thioesterase subunit) in Actinobacteria suggest that these COGs are less crucial to tryptophan biosynthesis. As elaborated in lines 330 to 335 of our article's updated version, the absence of the enzyme phosphoribosyl anthranilate isomerase TrpF (COG0135) has been compensated by the enzyme PirA, which provides the necessary isomerase activity.

The significance of the gradient color scale is also highlighted in the third example discussed in our article, focusing on key proteins i

nvolved in the eukaryotic transcription initiation process. As illustrated in Fig 4, cells representing COGs belonging to the first COG groups in all eukaryotic phyla are depicted in dark brown, indicating the presence of proteins crucial to this process. In contrast, cells corresponding to COGs less relevant to the eukaryotic transcription initiation process are depicted in yellow-like colors. Notably, these COGs are predominantly absent in ancient phyla, positioned close to the root of the phylogenetic tree that divides Bacteria, Archaea, and Eukarya (Fig 4). This suggests that only the more relevant proteins in the process are consistently shared among all eukaryotic phyla.

 In the updated version of the PhyloString web server, we have adjusted the color palette's scaling to enhance the visibility of distinctions among these values. Additionally, we have incorporated a practical example within the Tutorial web page to demonstrate how this color-coded information can aid in interpreting the analysis.

4/ A lot of thresholds are used without justification. What would happen if we would vary the a bit?

In response to the reviewer's comment, we have introduced a new feature on our PhyloString web server page, enabling users to vary and customize the parameter values according to their preferences. This new feature can be found selecting the Advance analysis option of PhyloString web page, as indicated in lines 175 to 176 of our revised version of the article. The pre-compiled analysis for predefined default values, which we have used as reference points, allows for quick output display and a rapid overview of various results. Subsequently, if users wish to fine-tune their analysis using values different from our predefined settings, the heatmaps and enrichment statistics will be generated dynamically in real time.

What would change if I change the query COG but I stay in the same functional set of proteins? How are the proteins in x-axis identified? Does the STRING index cutoff play a role in this selection? 

The inquiry about the potential outcomes when substituting the COG seed with another COG in the same functional category and whether the String index cutoff value may impact these outcomes is an insightful question. To provide a comprehensive answer, we have integrated it into the PhyloString Tutorial web page, accompanied by illustrative examples that highlight these scenarios. Tip 3 of this Tutorial web page demonstrates the difference observed when using KOG1932 (TATA-binding protein-associated factor) and KOG3219 (Transcription initiation factor TFIID, subunit TAF11) as query seed KOGs. This showcases that, while the results are generally comparable, they are significantly influenced by the coherence of the functional relationships within the study process and the centrality of the chosen COG as a query seed.

 Regarding the identification of proteins and phylogeny classes within the heatmap cells, as mentioned in our second response, we have transitioned from our previous static HTML heatmaps to a dynamic interactive output where relevant information about each cell is displayed when users hover their mouse cursor over each cell. This information includes the phylogenetic group, the COG or KOG ID with its description, the count of organisms considered in the specific cell, and its corresponding score, as described in the updated article, lines 273 to 277.

What would happen if I have 2 cells next to each other, one yellow and one red, and behind the yellow one the support for presence is 0.45 in the phylum, while it is 0.55 in the other cell? Is it really pertinent to put 0 and 1 which such similar proportions?

We acknowledge the reviewer's concerns regarding using a single cutoff value of 50% to determine the presence or absence of a family of orthologous genes (COG/KOG) within a phylogenetic group in the heatmaps generated by PhyloString. We agree that this approach can lead to contiguous cells exhibiting slight variations in their relative COG frequencies, resulting in different presence and absence assignments. However, we would like to emphasize that while this criterion may have limitations in representing relative frequencies close to 50%, it generally serves as a useful descriptor. Initially, we also considered assigning cells within the heatmap to represent the relative frequency of organisms with the presence of the COG within each phylogenetic group. Although this approach might seem more precise in theory, it often resulted in noisy and biologically challenging interpretations. However, given the large number of cells in the heatmap generated by PhyloString, the simplification of considering cells with relative frequencies greater than 50% as present proved to be the most practical and interpretable choice in most cases. We believe that the examples presented in the article and the new help and tutorial web pages support our decision and help users understand the rationale behind it.

5/ The webserver lacks documentation. It is easy to use but it is not so easy to understand the output without any tutorial or description.

As previously addressed in our responses to reviewer #1's second and third questions, the updated version of PhyloString incorporates several enhancements. We have introduced a Help page to provide users with an overview of the PhyloString web server, elucidate the input parameters required for their studies, delineate the various sections in the analysis output, and offer guidance on effectively interpreting the results.

In addition to the Help page, our PhyloString web server now includes a Tutorial web page strategically designed to lead users through the essential steps of a standard study. This tutorial also includes invaluable tips for harnessing PhyloString capabilities to their fullest and making sense of the diverse range of potential outcomes.

Finally, in the updated version of the article, we have included a new figure (Fig 1) with a schematic representation of the primary steps employed by PhyloString

Minor points:

- the figure legends should be more detailed.

Considering the reviewer's comment, the revised version of the article now provides more comprehensive and detailed legends for the figures, offering additional elements to facilitate their interpretation.

Response to Reviewer #3:

Reviewer #3: Dorantes-Torres et al. present a Web server, PhyloString, that analyses data, ingested from the STRING database and decomposes the associations in two-dimensional clustering, along the COG composition and the phylogenetic axes. The clustering is accompanied by a statistical enrichment/depletion estimate, based on an extremely naïve random independent combinatorial model. The server exists and serves the information claimed by the authors; this, technically, gives the whole enterprise a passing score for publication.

 The utility of the analysis is doubtful, given the primitive statistics and numerous ad hoc decisions, implemented in the pipeline (e.g. the binary presence/absence scoring at 50% of occurrence among the genomes in a phylogroup).

As previously discussed in our responses to reviewer #2's fourth question, although the binary scoring at the 50% occurrence threshold among genomes within a phylogenetic group may have limitations in representing the presence/absence tendencies of some COGs, it generally serves as a valuable descriptor. Initially, we explored the option of assigning cells within the heatmap to represent the relative frequency of each phylogenetic group to have organisms coding the corresponding COGs. However, this approach, although theoretically more precise, often led to noisy and biologically complex interpretations.

Given the substantial number of cells in the heatmap generated by PhyloString, we found that simplifying the representation by considering cells with relative frequencies greater than 50% as "present" was the most pragmatic and interpretable choice in most instances. We believe that the examples provided in the article and the new help and tutorial web pages offer ample support for our decision and help users grasp the reasoning behind it.

Using a hopelessly outdated COG annotation doesn't help either (e.g. CRISPR-Cas9 is annotated as "Uncharacterized protein conserved in bacteria", as it was in 2003 version). 

In response to the referee's feedback, we have implemented significant updates to our PhyloString server. The server now operates using the latest version, 12.0, of the STRING database. Furthermore, we have generated revised versions of each analysis presented in our article, demonstrating the application of PhyloString in exploring biological processes and metabolic pathways considering the updated STRING 12.0 score values. 

The authors make no attempt to find the data-driven granularity for clustering, leaving the user to experiment with the settings.

As we mentioned in our response to the fourth question raised by reviewer #2 and in consideration of the referee's comment, we have implemented a new feature on our PhyloString web server page. This feature empowers users to customize parameter values based on their preferences, allowing data-driven granularity for clustering, and facilitating the user to experiment with different settings. This new feature can be found selecting the Advanced analysis option of PhyloString web page, as indicated in lines 175 to 176 of our revised version of the article.

The pre-compiled analysis with predefined default values, which we have established as reference points, enables rapid output display and a quick overview of various results. Subsequently, should users want to fine-tune their analysis using values different from our predefined settings, the heatmaps and enrichment statistics will be dynamically generated in real-time. Our newly added Help and Tutorial pages provide examples demonstrating how to leverage data-driven granularity for clustering.

There is one problem that must be addressed in the revision: in the Methods section the authors mention using the R functions for hypergeometric distribution, but give no further details for the analysis they perform. Likewise, they use clustering facilities of the heatmap library without any reference to how the rows and columns are clustered. A detailed description of the analysis from the subject matter perspective (what is being computed and why, as opposed to details on how R functions are invoked), including the underlying assumptions and an assessment of their applicability to the real data, is necessary.

In response to the reviewer's request, we have included a detailed description of the statistical principles underpinning our analysis in the Materials and Methods Section, lines 124 to 224. This description includes the new options we recently implemented to allow the users to select different Clustering methods and Distance matrices employed in the clustering procedure. These Methods and Distance matrices are now described in the supplementary Tables S1 and S2 of the updated version of the article and in the new PhyloString Help web page.

---

## [Decision Letter · Decision Letter 1]

28 Dec 2023

PhyloString: a web server designed to identify, visualize, and evaluate functional relationships between orthologous protein groups across different phylogenetic lineages

PONE-D-23-24307R1

Dear Dr. Merino,

We’re pleased to inform you that your manuscript has been judged scientifically suitable for publication and will be formally accepted for publication once it meets all outstanding technical requirements.

Kind regards,

Shailender Kumar Verma, Ph.D.

Academic Editor

PLOS ONE

Additional Editor Comments (optional):

Reviewers' comments:

Reviewer's Responses to Questions

**Comments to the Author**

1. If the authors have adequately addressed your comments raised in a previous round of review and you feel that this manuscript is now acceptable for publication, you may indicate that here to bypass the “Comments to the Author” section, enter your conflict of interest statement in the “Confidential to Editor” section, and submit your "Accept" recommendation.

Reviewer #2: All comments have been addressed

Reviewer #3: (No Response)

2. Is the manuscript technically sound, and do the data support the conclusions?

Reviewer #2: Yes

Reviewer #3: Partly

3. Has the statistical analysis been performed appropriately and rigorously? 

Reviewer #2: Yes

Reviewer #3: N/A

4. Have the authors made all data underlying the findings in their manuscript fully available?

Reviewer #2: Yes

Reviewer #3: Yes

5. Is the manuscript presented in an intelligible fashion and written in standard English?

Reviewer #2: Yes

Reviewer #3: Yes

6. Review Comments to the Author

Reviewer #2: The authors did a very good job at addressing my concerns. They clarified some methodological points and they showcased the web server usefulness on several practical case studies.

Reviewer #3: The authors updated the COG annotations and provided better explanations to the algorithm details. The authors did not change the arbitrary binarization of the phyletic pattern and have not attempted to find the natural clustering thresholds (giving the users more clustering options is the opposite of the finding the data-driven granularity, as it shifts the burden on the user to explore and judge).

In my opinion this undermines the value of the tool, but it shouldn't prevent the publication of its description.

7. PLOS authors have the option to publish the peer review history of their article (what does this mean?). If published, this will include your full peer review and any attached files.

Reviewer #2: No

Reviewer #3: No

---

## [Editor Report · Acceptance letter]

18 Jan 2024

PONE-D-23-24307R1 

PLOS ONE

Dear Dr. Merino, 

I'm pleased to inform you that your manuscript has been deemed suitable for publication in PLOS ONE. Congratulations! Your manuscript is now being handed over to our production team.

Kind regards, 

on behalf of

Dr. Shailender Kumar Verma 

Academic Editor

PLOS ONE